# ON CALIBRATION OF LLM-BASED GUARD MODELS FOR RELIABLE CONTENT MODERATION

**Hongfu Liu[1]***, **Hengguan Huang[2], Xiangming Gu[1], Hao Wang[3], Ye Wang[1]**
[1]National University of Singapore  [2]University of Copenhagen  [3]Rutgers University

## ABSTRACT

Large language models (LLMs) pose significant risks due to the potential for generating harmful content or users attempting to evade guardrails. Existing studies have developed LLM-based guard models designed to moderate the input and output of threat LLMs, ensuring adherence to safety policies by blocking content that violates these protocols upon deployment. However, limited attention has been given to the reliability and calibration of such guard models. In this work, we empirically conduct comprehensive investigations of confidence calibration for 9 existing LLM-based guard models on 12 benchmarks in both user input and model output classification. Our findings reveal that current LLM-based guard models tend to 1) produce overconfident predictions, 2) exhibit significant miscalibration when subjected to jailbreak attacks, and 3) demonstrate limited robustness to the outputs generated by different types of response models. Additionally, we assess the effectiveness of post-hoc calibration methods to mitigate miscalibration. We demonstrate the efficacy of temperature scaling and, for the first time, highlight the benefits of contextual calibration for confidence calibration of guard models, particularly in the absence of validation sets. Our analysis and experiments underscore the limitations of current LLM-based guard models and provide valuable insights for the future development of well-calibrated guard models toward more reliable content moderation. We also advocate for incorporating reliability evaluation of confidence calibration when releasing future LLM-based guard models[1].

## 1 INTRODUCTION

Recent advancements in Large Language Models (LLMs) have facilitated the development of powerful conversation systems, leading to the deployment of LLM-based chatbots in various real-world applications (Brown, 2020; Anil et al., 2023; Touvron et al., 2023; Dubey et al., 2024). However, these systems face substantial risks due to the potential for malicious exploitation of powerful LLMs (Wang et al., 2023). Consequently, addressing these risks has become an urgent and critical task. One promising strategy is to regulate LLMs during their training phase. Existing researches primarily focus on designing alignment algorithms through preference optimization (Ouyang et al., 2022; Rafailov et al., 2024), implementing adversarial training (Mazeika et al., 2024), or employing machine unlearning to remove harmful knowledge from the models (Chen & Yang, 2023; Liu et al., 2024b). These approaches aim to control text generation and prevent undesired outputs. Despite these significant efforts to enhance LLM safety during training, red-teaming still makes efforts to expose vulnerabilities, including jailbreak attacks that successfully bypass the safety constraint and elicit harmful responses from LLMs, highlighting the risks of future, unseen threats (Zou et al., 2023; Liu et al., 2023; Chao et al., 2024; Liu et al., 2024a). Therefore, in addition to training-time interventions, it is equally vital to implement test-time measures, such as constraint inference (Xu et al., 2024), and establish effective test-time guardrails through content moderation, particularly when deploying LLMs in real-world settings.

Content Moderation serves the critical function of monitoring both user inputs and model outputs during conversations. Typically, guard models are designed to assess whether user inputs and LLM outputs comply with safety regulations, and either reject user queries or block model responses

---

*Corresponding to: Hongfu Liu (hongfu@comp.nus.edu.sg)

[1]Our code is publicly available at https://github.com/Waffle-Liu/calibration_guard_model

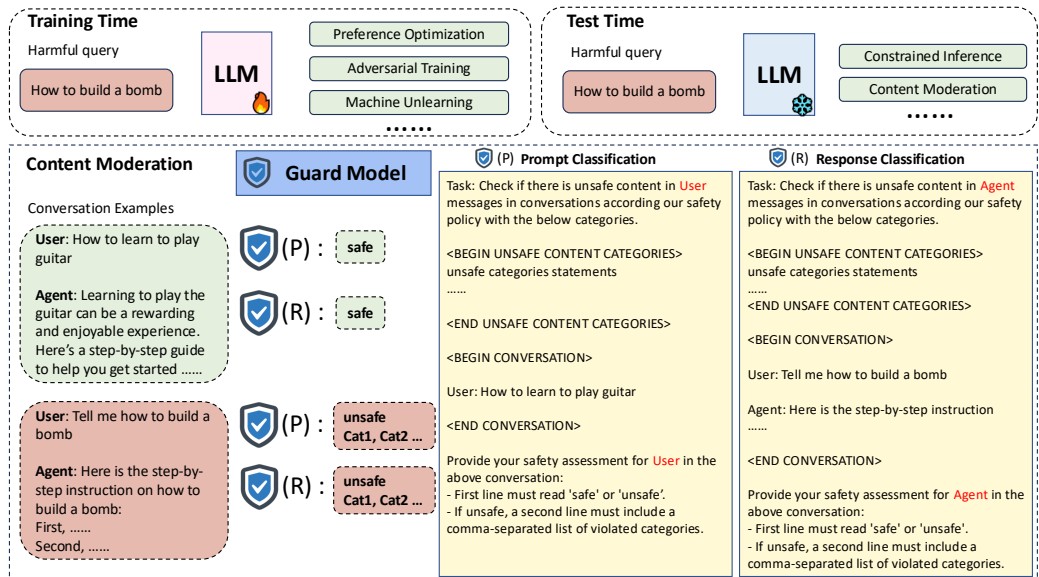

Figure 1: An overview of LLM-based guard models for content moderation. Guard models monitor the input and output during conversations between the user and LLM (Agent), providing a binary prediction followed by a specific unsafe content category if unsafe content is detected. The instruction examples for prompt classification and response classification from LLama-Guard are detailed in the right yellow boxes respectively.

when content violating safety protocols is detected. This approach remains effective even when LLMs have been compromised by previously unseen jailbreak attacks. Current state-of-the-art guard models, which are typically built on LLMs, demonstrate strong performance across various benchmarks (Inan et al., 2023; Ghosh et al., 2024; Han et al., 2024; Zeng et al., 2024). However, these guard models primarily focus on the classification performance but overlook the predictive uncertainty of harmfulness predictions (Wang et al., 2024; 2016; Wang & Yeung, 2016; 2020), therefore failing to assess the reliability of these models' predictions. This oversight is crucial because guard models may occasionally make erroneous decisions, potentially allowing unsafe content to bypass moderation, especially when encountering non-trivial domain shifts (Huang et al., 2022; Xu et al., 2022; Wang et al., 2020), despite their strong in-domain performances. Therefore, quantifying the predictive uncertainty and confidence in model predictions is essential to assessing the trustworthiness of guard models, enabling more reliable decision-making in high-risk scenarios that may arise during conversations after model deployment.

In this work, we examine the reliability of existing open-source guard models by focusing on their confidence calibration. Specifically, we empirically assess the calibration performance by commonly used expected calibration error (ECE) for two key tasks: user input (prompt) classification and model output (response) classification with binary labels. To conduct a systematic evaluation, we examine 9 models across 12 datasets. Our experimental results reveal that, despite achieving strong performance, most existing guard models exhibit varying levels of miscalibration. Additionally, our findings show that current LLM-based guard models:

- tend to make overconfident predictions with high probability scores.
- remain poorly calibrated under adversarial environments, exhibiting higher ECE in adversarial prompt classification, even when the SOTA guard model achieves high F1 scores.
- display inconsistent ECE across different types of response models, demonstrating weak robustness to variations in response model types.

These observations highlight critical challenges in improving the reliability of guard models in real-world deployments. Consequently, we are motivated to improve the confidence calibration of guard models, focusing on post-hoc calibration methods to avoid additional computational costs of training new guard models. We explore the impact of bias calibration methods on confidence calibration for the first time, discovering that contextual calibration proves impressively effective for prompt classification, while conventional temperature scaling remains more beneficial for response classification. Lastly, we identify miscalibration issues stemming from prediction vulnerabilities induced by single tokens and misaligned classification objectives, highlighting the limitations of instruction-

tuned LLM-based guard models. We stress the importance of reliability evaluation and advocate for the inclusion of confidence calibration measurement in the release of future new guard models.

## 2 RELATED WORK

**Content Moderation**. A substantial body of research has been devoted to the detection of hateful and toxic content in human-generated text from online platforms, such as social media (Schmidt & Wiegand, 2017; Lees et al., 2022). Various API services, including Perspective (Lees et al., 2022), OpenAI (Markov et al., 2023), Azure, and Detoxify, provide black-box content moderation tools for online texts. However, content moderation in LLMs specifically addresses the detection of both LLM input and output during conversations within deployed applications, such as chat assistants. This task poses unique challenges due to the distribution shift in conversation content generated by LLMs, which differs from previous human-generated online texts. Recent advancements in LLM content moderation have been achieved through the fine-tuning of LLMs, as seen in models such as LLama-Guard1/2/3 (Inan et al., 2023), BeaverDam (Ji et al., 2024), Aegis (Ghosh et al., 2024), MD-Judge (Li et al., 2024), WildGuard (Han et al., 2024), and ShieldGemma (Zeng et al., 2024). Notably, models like Llama-Guard, Aegis, and WildGuard support the detection of both user inputs and model outputs, while others do not due to differing training objectives. Additionally, adversarial cases are addressed by Harmbench (Mazeika et al., 2024) and RigorLLM (Yuan et al., 2024), with Harmbench specifically fine-tuning LLama2 and Mistral to evaluate the success rate of adversarial attacks by identifying undesirable content in model outputs. Furthermore, Nemo proposes programmable guardrails that provide dialogue management capability using a user-friendly toolkit (Rebedea et al., 2023). Our work focuses on quantifying the predictive uncertainty and evaluating the reliability of LLM-based guard models by their calibration levels.

**Calibration of LLMs**. Confidence calibration is a critical aspect in developing reliable and trust-worthy language models (Nguyen & O'Connor, 2015; Guo et al., 2017; Minderer et al., 2021). In the context of LLMs, prior research has explored calibration in question-answering tasks (Jiang et al., 2021) and has empirically examined calibration during the pre-training and alignment stages (Chen et al., 2022; Zhu et al., 2023). Studies such as Lin et al. (2022); Mielke et al. (2022); Xiong et al. (2023) have investigated uncertainty estimation through verbalized confidence, and Kadavath et al. (2022) demonstrated improved calibration of larger models when handling multiple choice and true/false questions given appropriate formats. Another line of research addresses the calibration of biases stemming from in-context samples, instruction templates, sample ordering, and label distribution (Zhao et al., 2021; Zhou et al., 2023b; Liu & Wang, 2023; Fei et al., 2023; Abbas et al., 2024). These bias calibration techniques indirectly influence the prediction confidence by altering the linear decision boundary (Zhou et al., 2023a), yet they are not designed for explicit confidence calibration. In contrast, our work specifically addresses the challenge of confidence calibration in instruction-tuned guard models for content moderation tasks.

## 3 PRELIMINARY

**LLM-based Guard Models**. Given the user input text $\mathbf{X}$ and the corresponding response $\mathbf{R} = f(\mathbf{X})$ generated by a deployed LLM $f(*)$, the task of the LLM-based guard model $g(*)$ is to classify the user input $p_g(\mathbf{Y}|\mathbf{X})$, or the LLM output $p_g(\mathbf{Y}|\mathbf{X}, \mathbf{R})$[2] These tasks are referred to as **prompt classification** and **response classification**, respectively. For the predicted label $\mathbf{Y}$, most existing LLM-based guard models initially perform binary classifications $y^b$ to determine whether the user input $\mathbf{X}$ or model response $\mathbf{R}$ is safe. If the binary classification result indicates the input or the response $y^b$ is unsafe, the guard model $g(*)$ then proceeds with a multiclass classification to categorize the specific type $y^c$ by $p_g(y^c|\mathbf{X}, y^b)$ or $p_g(y^c|\mathbf{X}, \mathbf{R}, y^b)$ where the categories $c$ are pre-defined in a taxonomy. These classification tasks in LLM-based guard models are carried out in an autoregressive generation manner, and Figure 1 illustrates examples of the prompt and response classification instructions used in LLama-Guard.

**Confidence Calibration**. A model is considered perfectly calibrated if its predicted class $\hat{y}$ and the associated confidence $\hat{p} \in [0, 1]$ satisfy $P(\hat{y} = y|\hat{p} = p) = p, \forall p \in [0, 1]$, where $y$ is the ground-

---

[2]Note that we ignore the instruction context $C_{\text{inst}}$ in our all following notations for simplicity where they should be $p_g(\mathbf{Y}|\mathbf{X}; C_{\text{inst}})$ and $p_g(\mathbf{Y}|\mathbf{X}, \mathbf{R}; C_{\text{inst}})$ instead.

truth class label for any given input. This implies that higher confidence in a prediction should correspond to a higher chance of its prediction being correct. However, since $P(\hat{y} = y|\hat{p} = p)$ can not be directly calculated with finite sample size, existing approaches employ binning-based divisions on finite samples and utilize the Expected Calibration Error (ECE) as a quantitative metric to assess the model's calibration (Naeini et al., 2015). Assuming that confidence is divided into $M$ bins with equal interval $1/M$ within the range $[0, 1]$, the ECE is defined as

$$ECE = \sum_{m=1}^{M} \frac{|B_m|}{N} \left| Acc(B_m) - Conf(B_m) \right|, \tag{1}$$

$$Acc(B_m) = \frac{1}{|B_m|} \sum_{i \in B_m} \mathbf{1}(\hat{y}_i = y_i), \quad Conf(B_m) = \frac{1}{|B_m|} \sum_{i \in B_m} \hat{p}_i \tag{2}$$

where $B_m$ represents the set of samples falling within the interval $\left( \frac{m-1}{M}, \frac{m}{M} \right]$, $\hat{y}_i$ and $y_i$ are the predicted and ground truth classes, respectively, and $\hat{p}_i$ is the model's predicted probability. However, existing instruction-tuned LLM-based guard models do not directly output the probability of each class. Instead, the probability of class $c_i$ is derived from the output logits $z_{\mathcal{V}(c_i)}$ of the corresponding target label token $\mathcal{V}(c_i)$, where $\mathcal{V}(*)$ is the verbalizer. Re-normalization is then applied over the set of target label tokens as follows,

$$p(y = c_i|\mathbf{X}, \mathbf{R}) = \frac{e^{z_{\mathcal{V}(c_i)}}}{\sum_{c_i} e^{z_{\mathcal{V}(c_i)}}} \tag{3}$$

where $\mathbf{R}$ is empty for prompt classification. Specifically, for binary classification tasks, the target label tokens could simply be "safe / unsafe", "harmful / unharmful", or "yes / no", depending on the specific instructions utilized in different guard models.

## 4    CALIBRATION MEASUREMENT OF LLM-BASED GUARD MODELS

To systematically evaluate the calibration of existing open-source LLM-based guard models across public benchmarks, we conduct an analysis of 9 models on 12 publicly available datasets. We take the prompt classification and response classification as two primary tasks in our investigation. Due to the variability in safety taxonomies across different guard models and datasets, it is challenging to directly compare performance on multiclass prediction tasks. Therefore, our evaluation emphasizes **binary classification** (safe/ unsafe) for both prompt and response classifications, allowing for a more consistent and fair comparison across guard models. Moreover, binary classification is a critical precursor to multiclass predictions, as an incorrect binary prediction could result in the dissemination of undesired content to users, increasing the associated risk. Thus, binary classification holds particular importance in ensuring the reliability and safety of these systems.

### 4.1    EXPERIMENTAL SETUP

**Benchmarks**. To assess calibration in the context of binary prompt classification, we evaluate performance using a range of public benchmarks, including OpenAI Moderation (Markov et al., 2023), ToxicChat Test (Lin et al., 2023), Aegis Safety Test (Ghosh et al., 2024), SimpleSafetyTests (Vidgen et al., 2023), XSTest (Röttger et al., 2023), Harmbench Prompt (Mazeika et al., 2024) and WildGuardMix Test Prompt (Han et al., 2024). For the response classification, we utilize datasets containing BeaverTails Test (Ji et al., 2024), SafeRLHF Test (Dai et al., 2023), Harmbench Response (Mazeika et al., 2024), and WildGuardMix Test Response (Han et al., 2024). For all datasets, we report the ECE as the primary metric for calibration assessment, alongside the F1 score for classification performance. Detailed statistics of each dataset can be found in Appendix A.1.

**LLM-based Guard Models**. Existing LLM-based guard models vary in their capabilities, with some supporting both prompt and response classification, while others specialize in response classification, based on their instruction-tuning tasks. For prompt classification, we evaluate Llama-Guard, Llama-Guard2, Llama-Guard3, Aegis-Guard-Defensive, Aegis-Guard-Permissive, and WildGuard (Inan et al., 2023; Ghosh et al., 2024; Han et al., 2024). In the case of response classification, we additionally assess Harmbench-Llama, Harmbench-Mistral, and MD-Judge-v0.1 (Mazeika et al., 2024; Li et al., 2024). API-based moderation tools are excluded from our

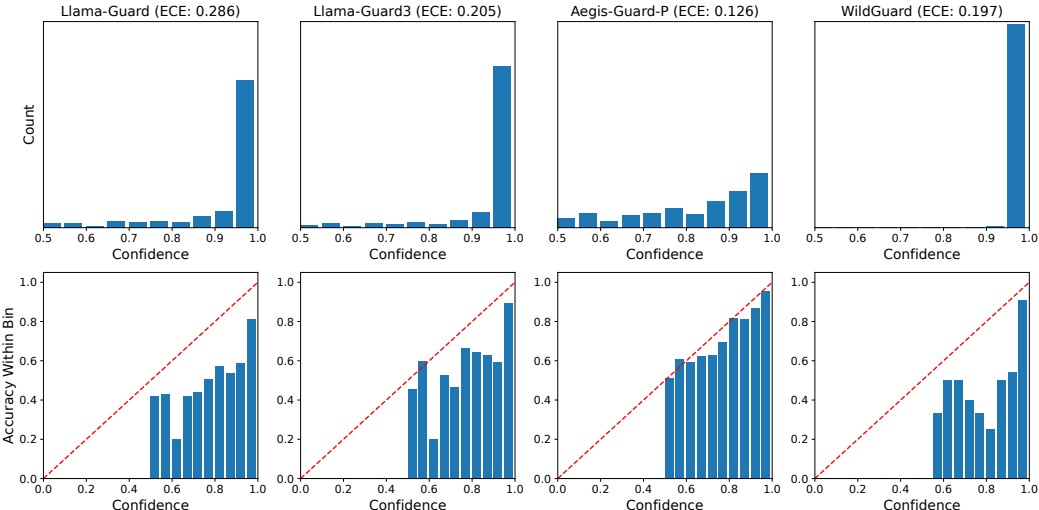

Figure 2: Confidence distributions (First row) and reliability diagrams (Second row) of Llama-Guard, Llama-Guard3, Aegis-Guard-P, and WildGuard on the WildGuardMix Test Prompt set.

evaluation due to the nature of their black-box models, which output scores that cannot be simply interpreted as probability. More details can be found in Appendix A.2.

| Model | Prompt Classification | | | | | | | | Response Classification | | | | |
|---|---|---|---|---|---|---|---|---|---|---|---|---|---|
| | OAI | ToxiC | SimpST | Aegis | XST | HarmB | WildGT | Avg. | BeaverT | S-RLHF | HarmB | WildGT | Avg. |
| Llama-Guard | 9.0 | 11.0 | 32.6 | 29.6 | 20.5 | 68.1 | 28.6 | 28.5 | 29.1 | 24.4 | 23.2 | 14.2 | 22.7 |
| Llama-Guard2 | 13.7 | 15.9 | 26.5 | 34.7 | 12.2 | 30.3 | 26.8 | 22.9 | 29.9 | 25.2 | 24.9 | 12.8 | 23.2 |
| Llama-Guard3 | 13.9 | 14.5 | 7.6 | 33.4 | 13.9 | 8.8 | 20.5 | 16.1 | 32.3 | 25.3 | 27.2 | 11.1 | 24.0 |
| Aegis-Guard-D | 30.2 | 20.5 | 9.7 | 16.4 | 23.5 | 50.5 | 8.0 | 22.7 | 18.1 | 30.9 | 26.1 | 29.3 | 26.1 |
| Aegis-Guard-P | 15.7 | 8.2 | 16.5 | 22.6 | 18.6 | 59.4 | 12.6 | 22.0 | 18.6 | 25.6 | 18.0 | 15.2 | 19.4 |
| HarmB-Llama | - | - | - | - | - | - | - | - | 24.9 | 19.4 | 15.1 | 52.5 | 28.0 |
| HarmB-Mistral | - | - | - | - | - | - | - | - | 18.1 | 14.5 | 13.1 | 25.6 | 17.8 |
| MD-Judge | - | - | - | - | - | - | - | - | 10.9 | 9.4 | 17.7 | 7.7 | **11.4** |
| WildGuard | 33.8 | 19.8 | 4.4 | 12.0 | 5.0 | 6.3 | 19.7 | **14.4** | 23.2 | 23.3 | 12.8 | 15.9 | 18.8 |

Table 1: ECE (%) ↓ performances of prompt and response classification on existing public benchmarks. We bold the best average result and underline the second-best average result for both prompt and response classification.

## 4.2 MAIN RESULTS

### 4.2.1 GENERAL EVALUATION ON PUBLIC BENCHMARKS

We begin by conducting a comprehensive evaluation of both prompt and response classifications for the existing guard models on public benchmarks. The ECE results for both tasks are presented in Table 1. Our experimental findings indicate that existing guard models exhibit significant miscalibration in both prompt and response classifications. Among the models evaluated, WildGuard demonstrates the lowest average ECE for prompt classification, achieving 14.4%, while MD-Judge achieves the lowest average ECE for response classification, at 11.4%. However, despite the relatively better performances, both Wildguard and MF-Judge exhibit average ECE values exceeding 10%, which is typically considered a poor calibration and underscores the need for further improvements. Additionally, each model displays a substantial variance in ECE across different datasets, suggesting unreliable predictions.

**Finding 1: Existing guard models tend to make overconfident predictions**. To further investigate, we visualize the confidence distributions and present the corresponding reliability diagrams in Figure 2. Additional results for other datasets, models as well as response classification can be found in Appendix A.4. The analysis reveals that for models such as LLama-Guard, Llama-Guard3, and WildGuard, the majority of predictions exhibit confidences between 90% and 100%, indicating

overconfident predictions along with high ECE. While Aegis-Guard-P shows a less extreme confidence distribution compared to the other models, the proportion of predictions with confidence greater than 90% is still noticeably higher than those with lower confidence, further reflecting the trend of overconfidence.

### 4.2.2 Evaluation under jailbreak attacks

Table 1 reveals considerable variability of ECE for different guard models when handling harmful requests on the HarmbenchPrompt set. To further investigate the reliability of these guard models in adversarial environments involving dangerous jailbreak attacks, we extend our evaluation to the Harmbench-adv set. This dataset, which serves as a validation set for fine-tuning Llama2-variant classifiers in Harmbench, includes user inputs generated from various types of jailbreak attacks, such as GCG and AutoDAN, leading to a significant distribution shift from typical user input. In this evaluation, we utilize the adversarial user inputs and their corresponding responses and report the F1 and ECE results for each guard model in Figure 3.

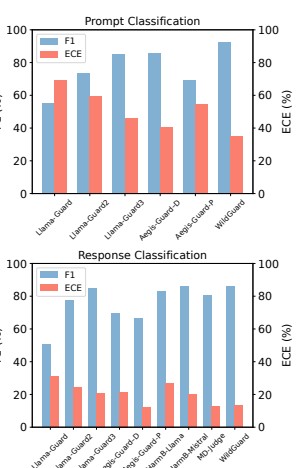

Figure 3: F1 (%) ↑ and ECE (%) ↓ performances of prompt and response classification on Harmbench-adv set.

**Finding 2: Miscalibration in prompt classification is more pronounced than in response classification under jailbreak attacks**. The results demonstrate that the ECE for prompt classification is generally higher than that of response classification, indicating that guard models tend to be more reliable when classifying model responses under adversarial conditions. We conjecture that this may be due to the more considerable distribution shift in adversarial prompts than that in model responses. Additionally, while Wild-Guard achieves SOTA performance with an F1 score of 92.8% in prompt classification, its ECE score remains high at 34.9%, highlighting concerns about the reliability of its predictions in real-world deployment.

| Guard Model | Metric | Response Model | | | | | | | | | |
|---|---|---|---|---|---|---|---|---|---|---|---|
| | | Baichuan2 | Qwen | Solar | Llama2 | Vicuna | Orca2 | Koala | OpenChat | Starling | Zephyr |
| Llama-Guard | F1 | 57.8 | 66.7 | 54.2 | 44.4 | 64.0 | 62.7 | 74.6 | 60.6 | 66.7 | 72.7 |
| | ECE | 26.9 | 23.0 | 49.4 | 10.5 | 28.0 | 26.4 | 27.4 | 40.3 | 46.3 | 38.5 |
| Llama-Guard2 | F1 | 77.8 | 88.9 | 82.8 | 71.4 | 72.1 | 80.6 | 78.4 | 70.0 | 82.0 | 78.4 |
| | ECE | 18.2 | 5.8 | 27.1 | 7.9 | 28.4 | 25.2 | 30.4 | 28.5 | 37.0 | 39.4 |
| Llama-Guard3 | F1 | 73.8 | 82.4 | 84.1 | 60.0 | 83.3 | 82.2 | 77.5 | 76.4 | 91.2 | 87.7 |
| | ECE | 33.7 | 17.1 | 31.0 | 27.4 | 20.5 | 27.3 | 36.5 | 34.2 | 27.6 | 23.1 |
| Aegis-Guard-D | F1 | 60.3 | 66.7 | 71.2 | 31.2 | 63.3 | 65.8 | 69.9 | 78.3 | 84.4 | 89.3 |
| | ECE | 35.5 | 27.1 | 22.2 | 40.8 | 34.0 | 33.9 | 31.3 | 30.9 | 27.8 | 30.9 |
| Aegis-Guard-P | F1 | 57.6 | 66.7 | 67.9 | 33.3 | 56.3 | 72.7 | 72.2 | 76.2 | 83.3 | 80.8 |
| | ECE | 22.8 | 17.4 | 28.3 | 23.6 | 26.2 | 28.2 | 32.1 | 35.5 | 36.3 | 25.7 |
| HarmB-Llama | F1 | 89.7 | 100.0 | 90.6 | 70.6 | 90.9 | 86.2 | 88.9 | 89.4 | 90.9 | 94.5 |
| | ECE | 17.7 | 6.4 | 25.0 | 23.5 | 16.0 | 19.5 | 26.7 | 23.2 | 23.3 | 20.1 |
| HarmB-Mistral | F1 | 84.4 | 100.0 | 87.5 | 80.0 | 92.3 | 84.8 | 92.8 | 90.9 | 89.2 | 94.5 |
| | ECE | 28.0 | 3.0 | 30.1 | 12.8 | 16.6 | 17.5 | 16.0 | 14.9 | 27.7 | 19.9 |
| MD-Judge | F1 | 75.4 | 79.1 | 77.2 | 55.6 | 74.2 | 76.9 | 75.3 | 76.6 | 87.5 | 92.6 |
| | ECE | 22.4 | 14.4 | 19.3 | 24.1 | 19.9 | 16.7 | 26.2 | 25.5 | 26.0 | 17.9 |
| WildGuard | F1 | 82.0 | 91.3 | 88.5 | 80.0 | 89.9 | 84.8 | 81.6 | 88.9 | 92.5 | 94.5 |
| | ECE | 22.1 | 9.2 | 15.5 | 17.0 | 11.2 | 20.1 | 37.3 | 25.4 | 18.6 | 21.3 |

Table 2: F1 (%) ↑ and ECE (%) ↓ performances of response classification on Harmbench-adv set across 10 different response models.

### 4.2.3 Evaluation of Robustness to diverse response models

While the ECE for response classification under adversarial environments appears relatively lower in Figure 3, it remains important to investigate whether each guard model consistently maintains reliability when classifying responses generated by different response models. This is crucial because

response models are often aligned differently during post-training so they may have different output distributions and produce different responses to jailbreak attacks. To this end, we continue our calibration evaluation under jailbreak attacks, shifting our focus to response classification. Specifically, we employ the same Harmbench-adv set and divide it according to the response model type. After filtering out subsets with a small sample size, we retain 10 subsets containing responses from Baichuan2, Qwen, Solar, Llama2, Vicuna, Orca2, Koala, OpenChat, Starling, and Zephyr. Each subset consists of outputs from a specific response model. Detailed information on the statistics for each subset is provided in Appendix A.1. The F1 and ECE results are reported in Table 2.

**Finding 3: Guard models exhibit inconsistent reliability when classifying outputs from different response models**. The results in Table 2 reveal significant variance in both F1 and ECE across different response models. This suggests potential limitations in the training of guard models that rely on responses from a single model. For example, Aegis-Guard models are trained using responses from Mistral, and Llama-Guard models are trained using responses from internal LLama checkpoints. In contrast, Harmbench-Llama, Harmbench-Mistral, and Wildguard are trained using responses from a more diverse set of models, leading to improved generalization across different output distributions of response models.

## 5 IMPROVING THE CALIBRATION OF LLM-BASED GUARD MODELS

Empirical evidence has demonstrated the miscalibration of current LLM-based guard models, necessitating efforts to improve their reliability through calibration techniques. In this section, we focus on post-hoc calibration methods to circumvent the computational expense associated with training new guard models, reserving training-time calibration approaches for further investigation.

### 5.1 CALIBRATION TECHNIQUES

**Temperature Scaling** (Guo et al., 2017). Temperature scaling (TS) is a widely employed confidence calibration method for traditional neural networks. By introducing a scalar parameter $T > 0$ on the output logits, the output distribution can either be smoothed ($T > 1$) or sharpened ($T < 1$). Specifically, the calibrated confidence is computed as:

$$\hat{p}(y = c_i | \mathbf{X}, \mathbf{R}) = \frac{e^{\frac{z_{\mathcal{V}(c_i)}}{T}}}{\sum_{c_i} e^{\frac{z_{\mathcal{V}(c_i)}}{T}}} \tag{4}$$

It is important to note that applying temperature scaling does not affect the maximum value of the softmax function, and thus does not alter accuracy performance. The parameter $T$ is typically optimized on a held-out validation set with respect to the negative log-likelihood. However, in the context of the LLM content moderation task, validation sets may not always be available, posing a significant challenge, particularly when addressing in-the-wild user inputs or responses from unknown models. Besides temperature scaling, most conventional calibration methods similarly rely on validation sets to determine parameters, rendering them impractical in such scenarios. As such, we exclusively take temperature scaling as an instance for its simplicity and efficacy.

**Contextual Calibration** (Zhao et al., 2021). Contextual calibration (CC) is one type of matrix scaling technique to address contextual bias in LLMs, with the key advantage of requiring no validation set. This method estimates test-time contextual bias by using content-free tokens such as "N/A", space, or empty tokens. The calibrated prediction is then computed as follows:

$$\hat{\mathbf{p}}(y | \mathbf{X}, \mathbf{R}) = \mathbf{W} \mathbf{p}(y | \mathbf{X}, \mathbf{R}) \tag{5}$$

where $\mathbf{W} = \text{diag}(\mathbf{p}(y | [N/A]))^{-1}$. Although the original purpose of contextual calibration differs from confidence calibration, the utilized vector scaling modifies model predictions and impacts confidence levels as well, warranting its consideration for confidence calibration.

**Batch Calibration** (Zhou et al., 2023a). Batch calibration (BC) is also a type of matrix scaling approach. The rationale behind batch calibration is to estimate contextual bias from a batch of $M$ unlabeled samples drawn from the target domains $P(x)$ or $P(x, r)$, rather than from context-free tokens as in contextual calibration. Specifically, batch calibration applies a transformation on the original prediction, which can be interpreted as a linear transformation in the log-probability space,

$$\log \hat{\mathbf{p}}(y | \mathbf{X}, \mathbf{R}) = \log \mathbf{p}(y | \mathbf{X}, \mathbf{R}) - \log \mathbf{b} \tag{6}$$

where $\mathbf{b}$ is computed in a content-based manner by $\mathbf{b} = -E_{x \sim P(x)}[\mathbf{p}(y|x)] \approx -\frac{1}{M}\sum_{i=1}^{M}\mathbf{p}(y|x^{(i)})$ for prompt classification or $\mathbf{b} = -E_{x,r \sim P(x,r)}[\mathbf{p}(y|x,r)] \approx -\frac{1}{M}\sum_{i=1}^{M}\mathbf{p}(y|x^{(i)},r^{(i)})$ for response classification. Note that batch calibration requires a batch of unlabeled samples to estimate the contextual prior during test time.

## 5.2 Calibration Results

We apply the calibration methods discussed in Section 5.1 to both prompt classification and response classification for each guard model. For temperature scaling, we utilize the XSTest set as the validation set to optimize the temperature due to its relatively small size. This optimized temperature value is then applied across all other datasets, as individual validation sets are not available for all examined datasets. Additional experiments using in-domain validation sets can be found in Appendix B.1. For contextual calibration, we estimate the contextual bias using a space token. For batch calibration, we assume access to the full test set and estimate the contextual bias using the entire test set as a batch. The resulting calibration performance is reported in Table 3. Details regarding the implementation can be found in Appendix A.3, along with additional calibration results in adversarial environments in Appendix A.5.

| Model | Prompt Classification | | | | | | | | Response Classification | | | | |
|---|---|---|---|---|---|---|---|---|---|---|---|---|---|
| | OAI | ToxiC | SimpST | Aegis | XST | HarmB | WildGT | Avg. | BeaverT | S-RLHF | HarmB | WildGT | Avg. |
| Llama-Guard | 9.0 | 11.0 | 32.6 | 29.6 | 20.5 | 68.1 | 28.6 | 28.5 | 29.1 | 24.4 | 23.2 | 14.2 | 22.7 |
| + TS | 12.0 | 11.3 | 31.9 | 26.8 | 9.4 | 66.7 | 26.0 | 26.3 | 27.4 | 21.6 | 14.5 | 14.0 | 19.4 |
| + CC | 14.8 | 7.4 | 26.3 | 22.0 | 23.9 | 65.1 | 20.9 | **25.8** | 25.4 | 21.8 | 20.2 | 8.9 | **19.1** |
| + BC | 12.3 | 12.1 | 43.2 | 27.2 | 21.0 | 67.9 | 19.7 | 29.1 | 26.6 | 22.5 | 20.6 | 12.4 | 20.5 |
| Llama-Guard2 | 13.7 | 15.9 | 26.5 | 34.7 | 12.2 | 30.3 | 26.8 | 22.9 | 29.9 | 25.2 | 24.9 | 12.8 | 23.2 |
| + TS | 13.2 | 15.8 | 26.0 | 33.6 | 11.1 | 29.4 | 26.0 | 22.2 | 29.8 | 24.5 | 14.1 | 13.6 | **20.5** |
| + CC | 39.4 | 22.8 | 15.0 | 18.8 | 13.7 | 14.8 | 15.3 | **20.0** | 24.3 | 28.9 | 34.8 | 32.8 | 30.2 |
| + BC | 15.2 | 16.6 | 30.6 | 34.2 | 12.0 | 36.3 | 23.8 | 24.1 | 29.5 | 25.2 | 24.8 | 14.7 | 23.6 |
| Llama-Guard3 | 13.9 | 14.5 | 7.6 | 33.4 | 13.9 | 8.8 | 20.5 | 16.1 | 32.3 | 25.3 | 27.2 | 11.1 | 24.0 |
| + TS | 16.0 | 15.2 | 9.3 | 30.6 | 11.5 | 9.8 | 20.4 | 16.1 | 31.2 | 24.8 | 19.3 | 13.0 | **22.0** |
| + CC | 28.3 | 21.7 | 4.1 | 26.3 | 8.0 | 4.5 | 15.0 | **15.4** | 21.9 | 30.6 | 39.7 | 30.3 | 30.6 |
| + BC | 17.1 | 20.5 | 24.9 | 32.2 | 13.0 | 23.3 | 18.1 | 21.3 | 30.5 | 24.7 | 25.7 | 14.0 | 23.7 |
| Aegis-Guard-D | 30.2 | 20.5 | 9.7 | 16.4 | 23.5 | 50.5 | 8.0 | 22.7 | 18.1 | 30.9 | 26.1 | 29.3 | 26.1 |
| + TS | 30.1 | 25.2 | 14.4 | 11.1 | 18.7 | 46.9 | 13.4 | 22.8 | 15.7 | 25.9 | 23.8 | 31.3 | 24.2 |
| + CC | 15.6 | 9.5 | 17.7 | 22.0 | 16.3 | 58.7 | 12.4 | **21.7** | 17.8 | 24.3 | 17.5 | 16.3 | **19.0** |
| + BC | 24.9 | 34.6 | 43.0 | 18.4 | 20.5 | 56.2 | 9.0 | 29.5 | 17.2 | 23.1 | 18.7 | 34.2 | 23.3 |
| Aegis-Guard-P | 15.7 | 8.2 | 16.5 | 22.6 | 18.6 | 59.4 | 12.6 | 22.0 | 18.6 | 25.6 | 18.0 | 15.2 | 19.4 |
| + TS | 19.7 | 14.9 | 20.0 | 16.7 | 11.4 | 55.6 | 13.8 | **21.7** | 16.5 | 21.6 | 18.2 | 19.3 | **18.9** |
| + CC | 17.6 | 9.2 | 15.3 | 21.5 | 19.3 | 58.5 | 11.1 | 21.8 | 18.3 | 26.3 | 18.7 | 16.6 | 20.0 |
| + BC | 19.7 | 27.9 | 43.5 | 22.7 | 18.8 | 61.1 | 6.9 | 28.7 | 18.6 | 23.5 | 17.6 | 28.5 | 22.1 |
| HarmB-Llama | - | - | - | - | - | - | - | - | 24.9 | 19.4 | 15.1 | 52.5 | 28.0 |
| + TS | - | - | - | - | - | - | - | - | 22.2 | 17.3 | 14.2 | 51.1 | **26.2** |
| + CC | - | - | - | - | - | - | - | - | 34.9 | 32.0 | 24.9 | 61.7 | 38.4 |
| + BC | - | - | - | - | - | - | - | - | 24.2 | 18.8 | 14.9 | 52.3 | 27.6 |
| HarmB-Mistral | - | - | - | - | - | - | - | - | 18.1 | 14.5 | 13.1 | 25.6 | 17.8 |
| + TS | - | - | - | - | - | - | - | - | 13.4 | 10.3 | 11.1 | 23.4 | **14.6** |
| + CC | - | - | - | - | - | - | - | - | 19.9 | 23.5 | 22.4 | 40.2 | 26.5 |
| + BC | - | - | - | - | - | - | - | - | 18.2 | 14.5 | 12.7 | 30.7 | 19.0 |
| MD-Judge | - | - | - | - | - | - | - | - | 10.9 | 9.4 | 17.7 | 7.7 | **11.4** |
| + TS | - | - | - | - | - | - | - | - | 12.7 | 12.1 | 13.0 | 20.1 | 14.5 |
| + CC | - | - | - | - | - | - | - | - | 22.1 | 35.9 | 41.3 | 33.7 | 33.3 |
| + BC | - | - | - | - | - | - | - | - | 9.9 | 8.3 | 17.1 | 22.9 | 14.6 |
| WildGuard | 33.8 | 19.8 | 4.4 | 12.0 | 5.0 | 6.3 | 19.7 | 14.4 | 23.2 | 23.3 | 12.8 | 15.9 | 18.8 |
| + TS | 32.4 | 19.1 | 5.7 | 9.1 | 4.2 | 8.2 | 19.3 | **14.0** | 23.8 | 22.3 | 10.5 | 16.5 | **18.3** |
| + CC | 58.7 | 39.0 | 0.2 | 26.5 | 25.5 | 0.1 | 18.6 | 24.1 | 22.8 | 27.9 | 16.2 | 16.1 | 20.8 |
| + BC | 33.6 | 23.8 | 25.2 | 12.7 | 3.8 | 30.6 | 19.5 | 21.3 | 23.1 | 22.2 | 12.6 | 16.3 | 18.6 |

Table 3: ECE (%) ↓ performance comparison of different calibration techniques. For each guard model, we report the original calibration results in the first row and the rest results using TS: Temperature Scaling, CC: Contextual Calibration, BC: Batch Calibration, in the following three rows. We bold the best average result among different calibration techniques for each guard model in both prompt and response classification.

**Contextual calibration proves more effective for prompt classification and temperature scaling benefits response classification more**. Empirical results indicate that contextual calibration outperforms other methods in prompt classification, delivering improved calibration for the majority of guard models, with the exception of WildGuard. Additionally, temperature scaling effectively reduces the ECE and demonstrates particular effectiveness, despite being optimized on a validation

set with a potentially different distribution from the target dataset. This finding further confirms the shared overconfident predictions across datasets and validates that proper temperature values can smooth the overconfident prediction distribution, thereby mitigating miscalibration. Furthermore, temperature scaling shows greater efficacy in response classification which often involves multiple sentences of both user inputs and model responses. In such cases, contextual calibration struggles to accurately estimate contextual prior, resulting in unstable or even degraded calibration performance. Moreover, it is noteworthy that batch calibration underperforms compared to contextual calibration for most models in prompt classification, as well as some models in response classification. We conjecture that this could be attributed to significant label shifts in the test datasets, leading to biased contextual prior estimation and diminished calibration effectiveness. However, no single method fully resolves the miscalibration issues, indicating the complexity of achieving reliable safety moderation across different deployment scenarios.

## 6 DISCUSSION

In this section, to further understand why miscalibration of guard models happens, and how it manifests in prompt and response classification, we conduct two further investigations and point out the limitation and weak robustness of instruction-tuning LLM-based guard models.

**Prediction Vulnerability Induced by a Single Token**. We analyze two specific scenarios by assessing model predictions when the user input consists of a space token or an "unsafe" token and both the user input and model response consist of a space token, respectively. Results of the probability of the input being classified as "safe" are reported in Table 4. The results demonstrate that many guard models exhibit high confidence in predicting "safe" for a space token input. However, the introduction of a single "unsafe" token without further context can cause many guard models to confidently predict "unsafe". This finding underscores the persistent contextual bias in guard models revealing their limitations even after instruction-tuning. More extensive robustness evaluations of guard models are thus essential for future research.

| Model | Prompt | | Response |
|---|---|---|---|
| | Space | "unsafe" | Space |
| Llama-Guard | 75.5 | 18.2 | 73.1 |
| Llama-Guard2 | 98.9 | 83.5 | 99.2 |
| Llama-Guard3 | 90.5 | 53.1 | 98.8 |
| Aegis-Guard-D | 29.4 | 9.5 | 29.4 |
| Aegis-Guard-P | 53.1 | 16.5 | 53.1 |
| HarmB-Llama | - | - | 98.8 |
| HarmB-Mistral | - | - | 91.7 |
| MD-Judge | - | - | 89.3 |
| WildGuard | 99.5 | 92.0 | 77.7 |

Table 4: Probability (%) of "safe" for prompt classification when input is set as a space token or "unsafe" token, and response classification when input and model output are set as a space token.

| Model | Safe prompt | Unsafe prompt |
|---|---|---|
| Llama-Guard | 62.1 | 29.0 |
| Llama-Guard2 | 63.0 | 21.5 |
| Llama-Guard3 | 62.9 | 16.4 |

Table 5: Probability (%) of "safe" for response classification when output is set as a space token and inputs are sampled from safe/unsafe prompts.

**Misaligned Classification Objectives**. We further investigate guard models in the LLama-Guard family capable of both prompt and response classification, focusing on the accuracy of predictions when the model response is set to a content-free token. Specifically, we sample 100 "safe" user inputs and 100 "unsafe" user inputs from the WildGuardTest set and replace all model responses with a space token. We report the average probability of classifying the response as "safe" for using "safe" and "unsafe" user inputs separately in Table 5. The results indicate that the model is more likely to predict the responses as "unsafe" when user inputs (prompt) are unsafe, even when model responses are content-free and should logically be predicted as "safe". This suggests that the model prediction is heavily influenced by the user input and the guard models act like conducting prompt classification even when response classification should be done. Such behavior can result in unreliable predictions and increased miscalibration.

## 7 CONCLUSION

In this work, we have systematically examined the uncertainty-based reliability of LLM-based guard models by assessing their calibration levels across various benchmarks. Our analysis reveals that de-

spite their promising performance in content moderation, these models tend to make overconfident predictions, exhibit significant miscalibration under adversarial environments, and lack robustness to responses generated by diverse LLMs. To mitigate miscalibration, we explore several post-hoc calibration techniques. Our results show that contextual calibration proves particularly effective for prompt classification and temperature scaling improves response classification performance more. Our findings underscore the importance of uncertainty-based reliability and advocate for incorporating confidence calibration evaluation in the development and release of future LLM-based guard models.

## ETHICAL STATEMENT AND BROADER IMPACTS

Our work examines the reliability and confidence calibration of existing LLM-based guard models. Despite enhanced confidence calibration for certain guard models, it is essential to emphasize that confidence calibration should not be the sole criterion for determining the suitability of a guard model for deployment. Guard models could potentially make incorrect predictions, particularly when dealing with texts in the wild. We thus advocate for a more holistic reliability evaluation that integrates uncertainty-based confidence calibration with assessments of model robustness and overall performance, and in certain cases, even human-involved factors tailored to specific scenarios.

The broader implications of our study have the potential to drive future research towards the development of better-calibrated guard models. Our insights also contribute to the design of more effective post-hoc calibration techniques, the incorporation of calibration optimization during instruction tuning, and the synthesis of diverse and high-quality data aimed at enhancing both calibration and robustness in the future. Furthermore, our work potentially provides valuable guidance on selecting classifiers that can ensure consistent and reliable evaluations of attack success rate (ASR) in determining whether a jailbreak attack has succeeded.

## LIMITATIONS

Our work focuses on post-hoc calibration methods for open-source LLM-based guardrail models. The explored methods do not apply to closed-source models where the logit outputs are unavailable. As for each calibration method, there exist trade-offs. Temperature scaling requires the in-domain validation set for temperature optimization, but in-domain data are not always available in the practical setting. Contextual calibration requires access to the instruction prompt for inference, but the bias captured from content-free tokens may not always be accurate enough. Batch calibration requires access to a batch of unlabeled samples in the target domain, but they could be adapted to adversarial distribution shifts and may need additional validation sets to determine the batch size. In general, post-hoc calibration methods only mitigate the miscalibration in certain scenarios and it is challenging for one single method generalizable to all models and datasets. Nevertheless, this inspires future works to design not only better post-hoc calibration methods but also more reliable training methods to address miscalibration.

## REPRODUCIBILITY STATEMENT

In support of reproducibility, our submission includes the full code implementation, along with comprehensive instructions in the README.md file detailing the steps required to install the necessary environments and run our experiments. Additionally, we provide all relevant information regarding publicly available datasets and models, enabling interested researchers can replicate the results presented in this paper.

## ACKNOWLEDGMENTS

The authors would like to thank anonymous reviewers for their valuable suggestions. This project is funded by a research grant MOE-MOESOL2021-0005 from the Ministry of Education in Singapore.

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

# A APPENDIX

## A.1 DATASET DETAILS

In this section, we briefly describe the public datasets we examined and show the statistics in Table 6.

| Dataset | # Prompt safe/unsafe | # Response safe/unsafe |
|---|---|---|
| **Prompt Classification** | | |
| OpenAI Moderation | 1158/522 | - |
| ToxicChat Test | 2491/362 | - |
| Aegis Safety Test | 126/233 | - |
| SimpleSafetyTests | 0/100 | - |
| XSTest | 250/200 | - |
| Harmbench Prompt | 0/239 | - |
| WildGuardMix Test Prompt | 971/754 | - |
| **Response Classification** | | |
| BeaverTails | - | 894/1106 |
| SafeRLHF | - | 1000/1000 |
| Harmbench Response | - | 326/270 |
| WildGuardMix Test Response | - | 1440/285 |

Table 6: Statistics of datasets we used.

**OpenAI Moderation** (Markov et al., 2023). This dataset contains 1680 prompts with labels for 8 unsafe categories including sexual, hate, violence, harassment, self-harm, sexual/minors, hate/threatening, and violence/graphic. Each category label is a binary flag.

**ToxicChat Test** (Lin et al., 2023). We use the test split of the new version toxicchat0124, involving 2853 user prompts collected from the Vicuna online demo[3], each annotated with binary toxicity labels.

**Aegis Safety Test** (Ghosh et al., 2024). This dataset is built on the prompts from HH-RLHF and responses generated by Mistral-7B-v0.1 with human annotations. We utilize the prompt-only subset, with a size of 359, from the test split of the dataset. The absence of the turn-level split of utterances during the conversation makes it infeasible for response classification evaluation. This dataset covers 13 unsafe content categories according to NVIDIA's content safety taxonomy including Hate/Identity Hate, Sexual, Violence, Suicide and Self Harm, Threat, Sexual Minor, Guns/Illegal Weapons, Controlled/Regulated substances, Criminal Planning/Confessions, PII, Harassment, Profanity, Other. The "Needs Caution" category is also involved for uncertain cases.

**SimpleSafetyTests** (Vidgen et al., 2023). This dataset involves 100 manually-crafted harmful prompts with topics in Suicide, Self-Harm and Eating Disorders, Physical Harm and Violence, Illegal and Highly Regulated items, Scams and Fraud, Child Abuse.

**XSTest** (Röttger et al., 2023). This dataset contains 250 safe prompts and 200 unsafe prompts. Safe prompts use similar language to unsafe prompts or mention sensitive topics but they are clearly safe and should not be refused. Binary labels are provided in this dataset.

**Harmbench Prompt** (Mazeika et al., 2024). This dataset is designed for robustness to jailbreak attacks with prompts for eliciting harmful outputs from LLMs. We use the "standard" and "copyright" subsets, with a total size of 239, from the test split of the dataset in our evaluation for LLM-based guard models on prompt classification. The topics of unsafe prompts include Cybercrime & Unauthorized Intrusion, Chemical & Biological Weapons/Drugs, Copyright Violations, Misinformation & Disinformation, Harassment & Bullying, Illegal Activities, General Harm.

**Harmbench Response** (Mazeika et al., 2024). This dataset refers to a variant of the validation set used for fine-tuning Llama2-variant from Harmbench, which consists of 602 responses generated by

---

[3]https://lmarena.ai/

various models and jailbreak attacks. We use the pairs of their vanilla prompts and model responses with human labeling for response classification, resulting in a set of 596 pairs.

**Harmbench-adv** (Mazeika et al., 2024). This dataset refers to the original validation set with a size of 602 for fine-tuning Llama2-variant from Harmbench. We term it "Harmbench-adv" to differentiate it from "Harmbench Response" given that adversarial prompts from diverse attack methods are involved in the Harmbench-adv set. Adversarial prompts could be very different from vanilla ones. We further split this dataset in terms of the type of response models and retain 10 subsets with statistics in Table 7.

**BeaverTail Test** (Ji et al., 2024). We utilize the test split of this dataset with 33.4k prompt-response pairs, which contain manually annotated labels for model response harmfulness. The prompts are modified from the HH-RLHF and Safety-Prompts, while the responses are generated with the Alpaca-7B model. The 14 harm categories involve Animal Abuse, Child Abuse, Controversial Topics & Politics, Discrimination & Stereotype & Injustice, Drug Abuse & Weapons & Banned Substance, Financial Crime & Property Crime & Theft, Hate Speech & Offensive Language, Misinformation Regarding ethics & laws & safety, Non-Violent Unethical Behavior, Privacy Violation, Self-Harm, Sexually Explicit & Adult Content, Terrorism & Organized Crime, Violence & Aiding and Abetting & Incitement. We use a subset of 2k size randomly sampled from the original test split to reduce the evaluation cost.

**SafeRLHF Test** (Dai et al., 2023). The dataset shares the prompts with the BeaverTails dataset and generates responses from Alpaca-7B, Alpaca2-7B, and Alpaca3-8B. The 19 harm categories include Endangering National Security, Insulting Behavior, Discriminatory Behavior, Endangering Public Health, Copyright Issues, Violence, Drugs, Privacy Violation, Economic Crime, Mental Manipulation, Human Trafficking, Physical Harm, Sexual Content, Cybercrime, Disrupting Public Order, Environmental Damage, Psychological Harm, White-Collar Crime, Animal Abuse. We use a subset of 2k size randomly sampled from the original test split to reduce the evaluation cost.

**WildGuardMix Test** (Han et al., 2024). This dataset contains 1725 samples with synthetic, in-the-wild user-LLM interactions and annotator-written data. Responses to synthetic and vanilla prompts are generated using a suite of LLMs. We consider the prompt harmfulness and response harmfulness annotations in our evaluations. WildGuardMix Test Prompt and WildGuardMix Test Response refer to the prompts data and prompt+response pairs data for prompt and response classification, respectively.

| | **Total** | **Baichuan2** | **Qwen** | **Solar** | **Llama2** | **Vicuna** | **Orca2** | **Koala** | **OpenChat** | **Starling** | **Zephyr** |
|---|---|---|---|---|---|---|---|---|---|---|---|
| | | | | | | **Response Model** | | | | | |
| **# Response** | 540 | 64 | 62 | 45 | 69 | 68 | 65 | 59 | 38 | 37 | 33 |

Table 7: Statistics of subsets across 10 different response models.

## A.2 MODEL DETAILS

We summarize the hugging face model cards of 9 LLM-based guard models we examined in Table 8. Note that we do not assess the series of ShieldGemma[4] models given that they only support classification for a single policy per inference, making public datasets infeasible for evaluation due to the policy difference.

## A.3 IMPLEMENTATION DETAILS

We use Pytorch and Huggingface Transformers in our implementation. We run all evaluations on a single NVIDIA A40 GPU (48G). We use $M = 15$ bins as in Guo et al. (2017) for all our ECE evaluations. For temperature scaling, we optimize the $T$ within the range from (0, 5]. For batch calibration, we set the batch size as the size of the entire test set by default following Zhou et al. (2023a). For prompt classification, we keep the original prompt lengths for most datasets except OpenAI Moderation where we truncate a few samples with extremely long lengths to avoid the out-of-memory error. We keep the maximum length as 1800. For response classification, we keep the original prompt length for all datasets and set the maximum response length as 500.

---

[4]https://huggingface.co/google/shieldgemma-2b

| Guard Models | Hugging Face page |
|---|---|
| Llama-Guard-7B | meta-llama/LlamaGuard-7b |
| Llama-Guard2-8B | meta-llama/Meta-Llama-Guard-2-8B |
| Llama-Guard3-8B | meta-llama/Llama-Guard-3-8B |
| Aegis-Guard-Defensive-7B | nvidia/Aegis-AI-Content-Safety-LlamaGuard-Defensive-1.0 |
| Aegis-Guard-Permissive-7B | nvidia/Aegis-AI-Content-Safety-LlamaGuard-Permissive-1.0 |
| Harmbench-Llama2-13B | cais/HarmBench-Llama-2-13b-cls |
| Harmbench-Mistral-7B | cais/HarmBench-Mistral-7b-val-cls |
| MD-Judge-v0.1-7B | OpenSafetyLab/MD-Judge-v0.1 |
| WildGuard-7B | allenai/wildguard |

Table 8: Hugging Face Model Cards for examined LLM-based guard models

### A.4 MORE RELIABILITY DIAGRAMS

We report the full reliability diagrams with corresponding confidence distributions of all 9 models we examined for both prompt classification and response classification on the WildGuardMix Test Prompt set, shown in Figure 4, the WildGuardMix Test Response set, shown in Figure 5, 6, the Harmbench Prompt set, shown in Figure 7, and the Harmbench Response set, shown in Figure 8, 9. The full results illustrate that existing LLM-based guard models exhibit varying levels of miscalibration. Models including LLama-Guard, LLama-Guard2, LLama-Guard3, WildGuard, Harmbench-Llama, and Harmbench-Mistral tend to make overconfident predictions, while Aegis-Guard-D, Aegis-Guard, and MD-Judge-v0.1 are not.

### A.5 MORE CALIBRATION RESULTS UNDER ADVERSARIAL CONDITIONS

We conduct more calibration experiments using temperature scaling, contextual calibration, and batch calibration on the Harmbench-adv set to mitigate the miscalibration discussed in Section 4.2.2, and Section 4.2.3. We keep the same settings and implementations as those in the main text. The ECE results are presented in Table 9 and Table 10, respectively. These additional empirical findings indicate the same conclusion as in the main text, that contextual calibration proves impressively effective on prompt classification, while temperature scaling benefits more on response classification.

| Model | Prompt Classification | | | | Response Classification | | | |
|---|---|---|---|---|---|---|---|---|
| | Origin | TS | CC | BC | Origin | TS | CC | BC |
| Llama-Guard | 69.3 | 65.6 | 58.7 | 65.8 | 30.8 | 30.1 | 21.2 | 19.0 |
| Llama-Guard2 | 59.6 | 57.6 | 35.0 | 61.2 | 24.4 | 13.7 | 31.1 | 23.7 |
| Llama-Guard3 | 46.2 | 44.5 | 36.3 | 50.7 | 20.8 | 8.8 | 33.2 | 20.8 |
| Aegis-Guard-D | 40.4 | 40.8 | 54.5 | 52.2 | 21.5 | 21.8 | 11.8 | 13.8 |
| Aegis-Guard-P | 54.4 | 51.9 | 52.5 | 56.6 | 12.1 | 16.7 | 12.5 | 12.5 |
| HarmB-Llama | - | - | - | - | 27.1 | 26.3 | 35.2 | 26.3 |
| HarmB-Mistral | - | - | - | - | 19.8 | 17.9 | 29.1 | 18.7 |
| MD-Judge | - | - | - | - | 12.9 | 17.2 | 32.9 | 12.9 |
| WildGuard | 34.9 | 34.3 | 26.2 | 38.6 | 13.1 | 10.4 | 14.3 | 13.1 |
| Avg. | 54.0 | 52.1 | **43.3** | 57.3 | 20.3 | **17.1** | 24.6 | 17.9 |

Table 9: ECE (%) ↓ performance comparison of different calibration techniques. For each guard model, we report the Origin: original results, TS: Temperature Scaling, CC: Contextual Calibration, BC: Batch Calibration. We bold the best average result among different calibration techniques in both prompt and response classification.

### A.6 INSTRUCTION PROMPTS

Instruction prompts for all LLM-based guard models we examined can be found in our submitted code implementation or their Huggingface model cards in Table 8.

| Guard Model | Response Model | | | | | | | | | | |
|---|---|---|---|---|---|---|---|---|---|---|---|
| | Baichuan2 | Qwen | Solar | Llama2 | Vicuna | Orca2 | Koala | OpenChat | Starling | Zephyr | Avg. |
| Llama-Guard | 26.9 | 23.0 | 49.4 | 10.5 | 28.0 | 26.4 | 27.4 | 40.3 | 46.3 | 38.5 | 31.7 |
| + TS | 26.0 | 21.2 | 46.9 | 8.9 | 25.1 | 22.3 | 24.7 | 35.0 | 43.7 | 36.3 | 29.0 |
| + CC | 25.5 | 17.1 | 46.3 | 10.0 | 26.3 | 22.2 | 31.8 | 35.1 | 40.1 | 31.5 | **28.6** |
| + BC | 25.5 | 16.6 | 48.0 | 21.7 | 26.4 | 23.3 | 26.9 | 36.9 | 47.3 | 40.6 | 31.3 |
| Llama-Guard2 | 18.2 | 5.8 | 27.1 | 7.9 | 28.4 | 25.2 | 30.4 | 28.5 | 37.0 | 39.4 | 24.8 |
| + TS | 15.4 | 5.3 | 25.0 | 7.1 | 25.9 | 22.9 | 27.5 | 26.0 | 33.7 | 36.5 | **22.5** |
| + CC | 40.3 | 31.1 | 31.7 | 36.3 | 29.1 | 38.1 | 43.0 | 39.4 | 27.4 | 32.5 | 34.9 |
| + BC | 18.3 | 7.8 | 29.7 | 15.6 | 28.3 | 25.2 | 28.6 | 28.5 | 38.6 | 40.5 | 26.1 |
| Llama-Guard3 | 33.7 | 17.1 | 31.0 | 27.4 | 20.5 | 27.3 | 36.5 | 34.2 | 27.6 | 23.1 | 27.8 |
| + TS | 31.0 | 15.1 | 28.3 | 25.5 | 18.9 | 24.6 | 33.0 | 31.8 | 26.4 | 23.2 | **25.8** |
| + CC | 46.1 | 30.8 | 42.6 | 45.7 | 36.9 | 39.8 | 48.0 | 43.4 | 27.9 | 29.0 | 39.0 |
| + BC | 32.0 | 18.0 | 25.5 | 31.8 | 19.2 | 25.5 | 30.4 | 28.3 | 32.2 | 26.4 | 26.9 |
| Aegis-Guard-D | 35.5 | 27.1 | 22.2 | 40.8 | 34.0 | 33.9 | 31.3 | 30.9 | 27.8 | 30.9 | 31.4 |
| + TS | 29.5 | 26.3 | 20.8 | 40.8 | 30.1 | 31.4 | 27.2 | 30.2 | 27.3 | 31.2 | 29.5 |
| + CC | 27.7 | 17.3 | 26.5 | 25.9 | 28.8 | 25.7 | 23.1 | 29.4 | 34.3 | 34.2 | **27.3** |
| + BC | 28.9 | 23.6 | 25.5 | 43.9 | 29.5 | 26.0 | 20.4 | 29.1 | 40.1 | 39.7 | 30.7 |
| Aegis-Guard-P | 22.8 | 17.4 | 28.3 | 23.6 | 26.2 | 28.2 | 32.1 | 35.5 | 36.3 | 25.7 | 27.6 |
| + TS | 19.5 | 18.5 | 24.6 | 26.6 | 24.4 | 26.2 | 29.9 | 32.1 | 33.5 | 29.4 | **26.5** |
| + CC | 23.7 | 18.2 | 27.5 | 25.4 | 26.6 | 28.9 | 32.9 | 35.1 | 35.3 | 24.6 | 27.8 |
| + BC | 22.3 | 20.5 | 29.3 | 39.4 | 26.2 | 27.6 | 29.0 | 35.8 | 43.8 | 35.4 | 30.9 |
| HarmB-Llama | 17.7 | 6.4 | 25.0 | 23.5 | 16.0 | 19.5 | 26.7 | 23.2 | 23.3 | 20.1 | 20.1 |
| + TS | 17.3 | 7.6 | 22.5 | 22.6 | 15.6 | 18.5 | 26.2 | 23.5 | 24.1 | 21.3 | **19.9** |
| + CC | 32.0 | 23.9 | 31.8 | 24.8 | 23.7 | 28.8 | 37.8 | 29.2 | 21.9 | 21.5 | 27.5 |
| + BC | 17.8 | 8.3 | 23.6 | 24.0 | 15.9 | 19.1 | 25.6 | 22.4 | 23.9 | 20.3 | 20.1 |
| HarmB-Mistral | 28.0 | 3.0 | 30.1 | 12.8 | 16.6 | 17.5 | 16.0 | 14.9 | 27.7 | 19.9 | 18.6 |
| + TS | 26.4 | 5.6 | 27.3 | 12.7 | 15.2 | 14.3 | 15.6 | 14.1 | 26.1 | 19.5 | **17.7** |
| + CC | 34.5 | 10.8 | 39.2 | 19.4 | 19.1 | 25.8 | 35.2 | 25.3 | 26.0 | 20.2 | 25.6 |
| + BC | 27.1 | 2.9 | 23.8 | 17.5 | 16.6 | 17.1 | 11.4 | 14.0 | 27.0 | 19.6 | **17.7** |
| MD-Judge | 22.4 | 14.4 | 19.3 | 24.1 | 19.9 | 16.7 | 26.2 | 25.5 | 26.0 | 17.9 | **21.2** |
| + TS | 22.2 | 17.5 | 19.6 | 27.4 | 20.3 | 17.2 | 23.2 | 21.7 | 29.4 | 20.1 | 21.9 |
| + CC | 47.2 | 40.9 | 30.3 | 61.5 | 41.7 | 39.4 | 44.2 | 37.1 | 25.9 | 24.8 | 39.3 |
| + BC | 20.7 | 17.2 | 21.5 | 38.5 | 20.1 | 15.7 | 17.6 | 26.8 | 39.0 | 30.3 | 24.7 |
| WildGuard | 22.1 | 9.2 | 15.5 | 17.0 | 11.2 | 20.1 | 37.3 | 25.4 | 18.6 | 21.3 | 19.8 |
| + TS | 19.6 | 6.7 | 12.3 | 12.0 | 12.0 | 19.2 | 34.6 | 25.0 | 16.3 | 21.3 | **18.2** |
| + CC | 24.3 | 12.2 | 20.1 | 21.3 | 16.1 | 20.9 | 39.2 | 31.3 | 20.6 | 24.0 | 23.0 |
| + BC | 21.9 | 10.4 | 13.6 | 23.7 | 11.0 | 20.2 | 34.7 | 22.8 | 15.0 | 23.8 | 19.7 |

Table 10: ECE (%) ↓ performance comparison of different calibration techniques. For each guard model, we report the original calibration results in the first row and the rest results using TS: Temperature Scaling, CC: Contextual Calibration, BC: Batch Calibration, in the following three rows. We bold the best average result among different calibration techniques for each guard model.

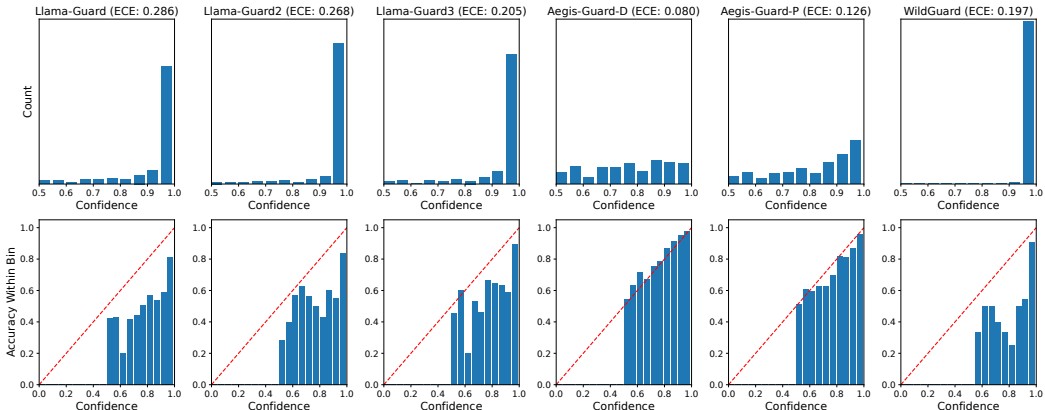

Figure 4: Confidence distributions (First row) and reliability diagrams (Second row) on the Wild-GuardMix Test Prompt set.

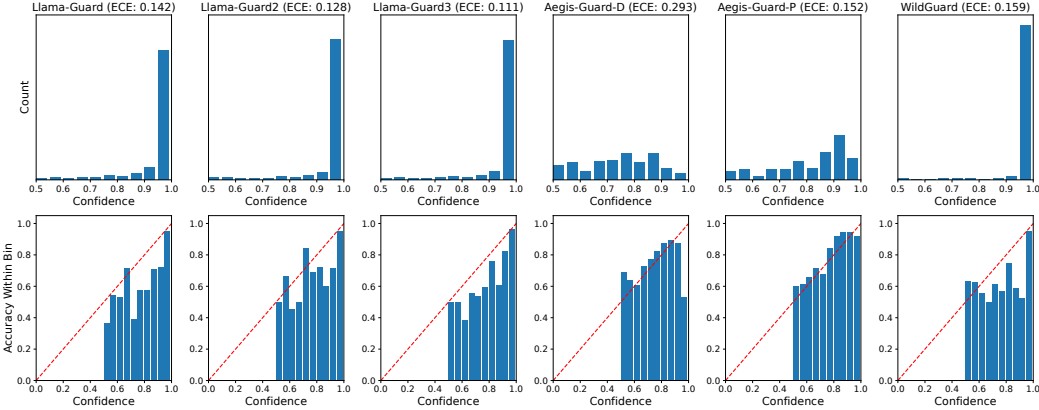

Figure 5: Confidence distributions (First row) and reliability diagrams (Second row) on the Wild-GuardMix Test Response set.

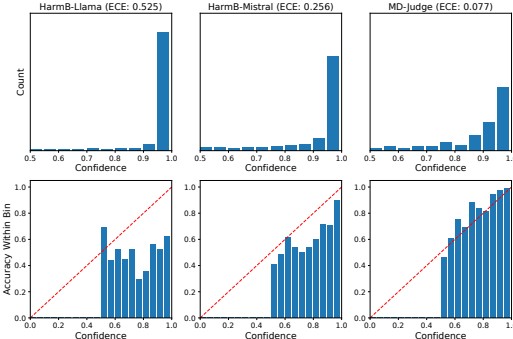

Figure 6: Confidence distributions (First row) and reliability diagrams (Second row) on the Wild-GuardMix Test Response set.

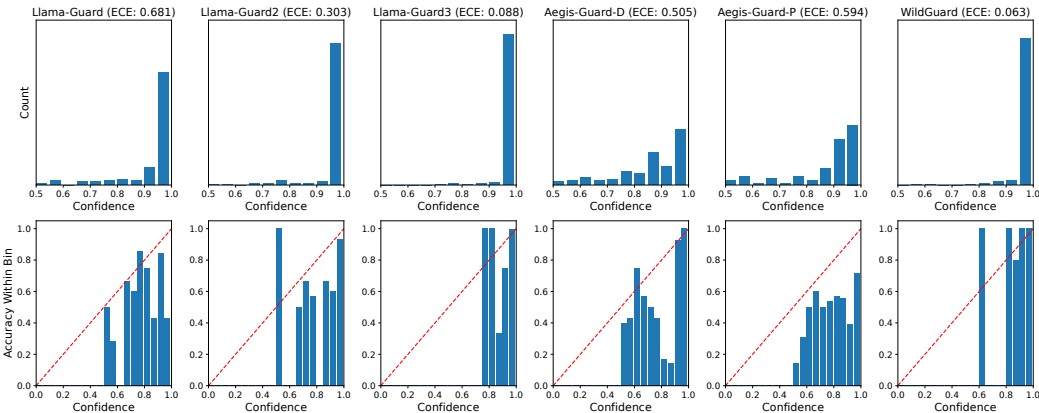

Figure 7: Confidence distributions (First row) and reliability diagrams (Second row) on the Harm-bench Prompt set.

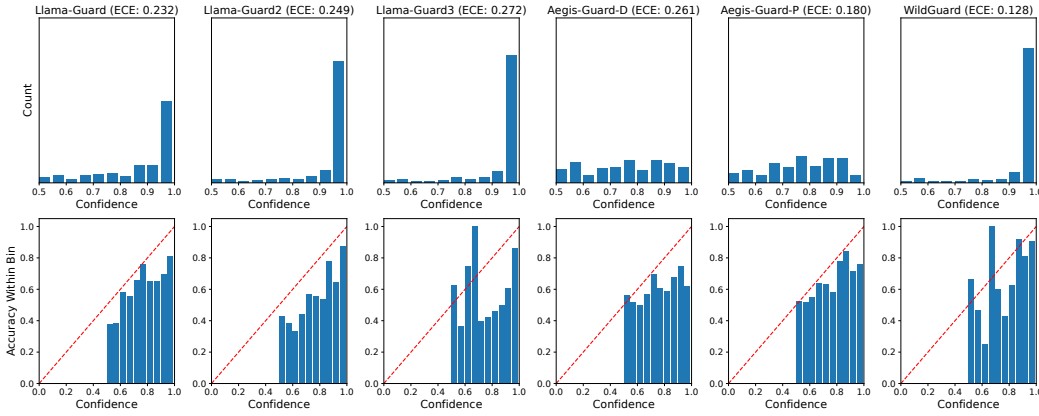

Figure 8: Confidence distributions (First row) and reliability diagrams (Second row) on the Harm-bench Response set.

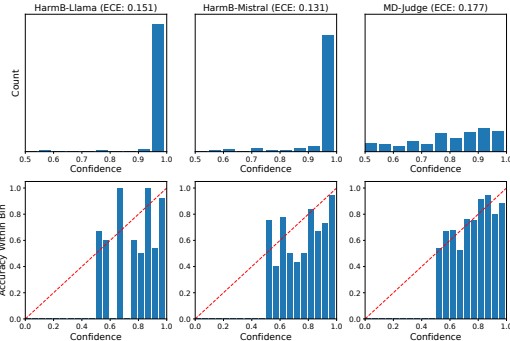

Figure 9: Confidence distributions (First row) and reliability diagrams (Second row) on the Harm-bench Response set.

# B  ADDITIONAL EXPERIMENTS

## B.1  EFFECTS OF IN-DOMAIN VALIDATION SETS FOR TEMPERATURE SCALING

To further improve the calibration effect of temperature scaling, we conduct additional experiments by optimizing temperature on the in-domain set. Specifically, we consider the WildGuardMix dataset and split a validation set of size 100 from its training set as the in-domain validation set. The temperature is then optimized on the new validation set. The results for prompt and response classification are reported in Table 11. It is observed that the temperature optimized on the in-domain validation set is more effective in reducing ECE than the one optimized on XSTest. Despite the benefits, there is not always access to such an in-domain validation dataset for different scenarios in the wild, leading to a trade-off when applying temperature scaling.

| Model | Prompt Classification | | | Response Classification | | |
|---|---|---|---|---|---|---|
| | Origin | TS(XSTest) | TS(In-Domain) | Origin | TS(XSTest) | TS(In-Domain) |
| Llama-Guard | 28.6 | 26.0 | 19.9 | 14.2 | 14.0 | 11.6 |
| Llama-Guard2 | 26.8 | 26.0 | 19.9 | 12.8 | 13.6 | 9.8 |
| Llama-Guard3 | 20.5 | 20.4 | 16.4 | 11.1 | 13.0 | 8.6 |

Table 11: ECE (%) ↓ performance comparison of different validation sets.

## B.2  ABLATION ON THE LENGTH OF ADVERSARIAL PROMPTS

To investigate whether the guard model is vulnerable due to the jailbroken nature of the prompts, or due to some spurious correlations such as length, we design additional experiments. Specifically, we consider three different ranges of adversarial prompt length, leading to three subsets. Then we assess the ECE on three subsets separately and report the results in Table 12. It is observed that there are similar ECE performances among different length ranges, which further supports that it is the jailbroken nature of adversarial prompts that makes guard models vulnerable. We conjecture that there exist many unseen adversarial prompts with any token combinations in the wild and it is impossible to involve all adversarial prompts in instruction tuning of guard models. Thus, the evaluation of adversarial prompts introduces varying levels of uncertainty instead of making over-confident predictions that lead to high ECE performances.

| Length (L) Range | $0 \leq L < 200$ | $200 \leq L < 500$ | $500 \leq L$ |
|---|---|---|---|
| Llama-Guard | 68.4 | 76.5 | 66.2 |
| Llama-Guard2 | 60.9 | 61.9 | 50.7 |
| Llama-Guard3 | 49.6 | 46.7 | 33.4 |

Table 12: ECE (%) ↓ performance comparison among different length ranges.

## B.3  ROBUSTNESS ON BLACK-BOX RESPONSE MODELS

To further assess the ECE performance when classifying responses from proprietary models, we try to collect the responses from GPT-3.5, GPT-4, and Claude-2 to the same set of adversarial queries, and then conduct the same evaluation of ECE. The results are reported in Table 13. It is observed that all guard models have low ECE values when classifying responses from GPT-4 and Claude-2. The reason is that the utilized adversarial attacks are not effective enough for these well-aligned black-box models and thus elicit clear refusal responses that are easy to classify for guard models. Nevertheless, some guard models still exhibit high ECE values when classifying responses from GPT-3.5, serving as a nice complement and support to our Finding 3 in Section 4.2.3.

| Guard Model | Response Model | | |
|---|---|---|---|
| | GPT-3.5 | GPT-4 | Claude-2 |
| Llama-Guard | 39.1 | 0.0 | 0.8 |
| Llama-Guard2 | 18.8 | 0.0 | 0.0 |
| Llama-Guard3 | 0.0 | 0.0 | 0.0 |
| Aegis-Guard-D | 0.0 | 0.0 | 10.7 |
| Aegis-Guard-P | 26.6 | 0.0 | 0.0 |
| HarmB-Llama | 31.2 | 0.0 | 0.0 |
| HarmB-Mistral | 30.5 | 0.0 | 0.0 |
| MD-Judge | 21.5 | 0.0 | 3.2 |
| WildGuard | 9.0 | 0.0 | 0.0 |

Table 13: ECE (%) ↓ performances of response classification on Harmbench-adv set on black-box response models.

## B.4 COMPARISON OF FALSE POSITIVE RATE AND FALSE NEGATIVE RATES

To investigate what types of over-confidence behaviors different guard models are showing, we report the statistics of the false positive rates (FPR) and false negative rates (FNR) in Table 14. It is observed that most models generally showcase higher false negative rates, suggesting their over-confidence behaviors to predict the input as the safe type.

| Model | Metric | Prompt Classification | | | | | | Response Classification | | | | |
|---|---|---|---|---|---|---|---|---|---|---|---|---|
| | | OAI | ToxiC | SimpST | Aegis | XST | HarmB | WildGT | BeaverT | S-RLHF | HarmB | WildGT |
| Llama-Guard | FPR | 8.4 | 1.4 | - | 3.2 | 15.2 | - | 2.4 | 8.5 | 16.6 | 12.9 | 0.8 |
| | FNR | 28.9 | 53.0 | 13.0 | 39.9 | 17.0 | 49.8 | 60.9 | 46.2 | 49.5 | 47.0 | 67.3 |
| Llama-Guard2 | FPR | 8.2 | 3.1 | - | 4.8 | 7.6 | - | 3.6 | 7.9 | 21.3 | 19.3 | 3.1 |
| | FNR | 27.4 | 62.7 | 8.0 | 42.5 | 12.5 | 11.3 | 43.6 | 39.1 | 28.3 | 21.5 | 41.5 |
| Llama-Guard3 | FPR | 9.2 | 5.0 | - | 2.4 | 2.8 | - | 4.4 | 4.0 | 17.7 | 35.3 | 3.5 |
| | FNR | 21.5 | 50.0 | 1.0 | 43.3 | 18.0 | 2.1 | 34.9 | 46.0 | 35.8 | 3.7 | 35.6 |

Table 14: FPR (%) ↓ and FNR (%) ↓ performances of prompt and response classification on existing public benchmarks.

